# Breast Cancer: Extracellular Matrix and Microbiome Interactions

**DOI:** 10.3390/ijms25137226

**Published:** 2024-06-30

**Authors:** Lourdes Herrera-Quintana, Héctor Vázquez-Lorente, Julio Plaza-Diaz

**Affiliations:** 1Department of Physiology, Schools of Pharmacy and Medicine, University of Granada, 18071 Granada, Spain; lourdesherrera@ugr.es (L.H.-Q.); hectorvazquez@ugr.es (H.V.-L.); 2Biomedical Research Center, Health Sciences Technology Park, University of Granada, 18016 Granada, Spain; 3Department of Biochemistry and Molecular Biology II, School of Pharmacy, University of Granada, Campus de Cartuja s/n, 18071 Granada, Spain; 4Instituto de Investigación Biosanitaria IBS.GRANADA, Complejo Hospitalario Universitario de Granada, 18014 Granada, Spain; 5Children’s Hospital of Eastern Ontario Research Institute, Ottawa, ON K1H 8L1, Canada

**Keywords:** breast cancer, extracellular matrix, microbiome, tumorigenesis

## Abstract

Breast cancer represents the most prevalent form of cancer and the leading cause of cancer-related mortality among females worldwide. It has been reported that several risk factors contribute to the appearance and progression of this disease. Despite the advancements in breast cancer treatment, a significant portion of patients with distant metastases still experiences no cure. The extracellular matrix represents a potential target for enhanced serum biomarkers in breast cancer. Furthermore, extracellular matrix degradation and epithelial–mesenchymal transition constitute the primary stages of local invasion during tumorigenesis. Additionally, the microbiome has a potential influence on diverse physiological processes. It is emerging that microbial dysbiosis is a significant element in the development and progression of various cancers, including breast cancer. Thus, a better understanding of extracellular matrix and microbiome interactions could provide novel alternatives to breast cancer treatment and management. In this review, we summarize the current evidence regarding the intricate relationship between breast cancer with the extracellular matrix and the microbiome. We discuss the arising associations and future perspectives in this field.

## 1. Introduction

Breast cancer (BC) is a severe, life-threatening condition widely recognized as a prevalent contributor to cancer mortality, being the most prevalent malignancy afflicting women [1]. Despite significant advancements in its diagnosis and treatment, BC metastasis continues to be a leading cause of mortality among women, causing an imperative need for a deeper understanding of the molecular and cellular mechanisms that empower cancer cells to not only survive but also expand [2]. Several risk factors have been described in the appearance and progression of this disease, including genetic factors (such as inherited mutations in genes), reproductive factors (e.g., age at menarche or age at first live birth), high breast density, dietary patterns, alcohol consumption, sedentary lifestyle, or obesity [3,4,5]. In this complex scenario, preclinical and clinical data have accumulated evidence indicating that some of these risk factors may worsen the incidence, severity, and mortality of many types of cancer, including BC, due to its relationship with adipose tissue-related chronic inflammation, immune cell dysregulation, angiogenesis, extracellular matrix (ECM) stiffness, and genomic instability [6].

Although conventional treatments such as chemotherapy, surgery, hormone therapy, or radiotherapy are commonly used, they often entail significant side effects. Therefore, there is a critical need to investigate more cost-effective and efficient treatment modalities for BC [7] like personalized medicine [8] based on genetic, phenotypic, and environmental factors, minimally invasive procedures [9] such as laparoscopic and robotic surgery, improved screening and early detection [10], cost-effective [11] drug development, combination therapies [12] using combinations of drugs that work synergistically, and telemedicine and remote monitoring [13].

A novel factor that has emerged in recent years and that could be linked with BC is the microbiome, which has been identified as a fundamental contributor to human health and overall wellbeing [14]. It can exert influence on a multitude of human biological, hormonal, and metabolic pathways. This influence can potentially stimulate the initiation, proliferation, and genetic instability of cancer within host cells or, alternatively, induce apoptosis [15]. The interplay between tumor cells and their microenvironment has been recognized as one of the key determinants of cancer development; data suggest that both the ECM and the microbiome represent microenvironments that contribute to the onset and progression of tumors [16]. ECM turnover represents a potential target for novel and enhanced serum biomarkers due to ECM homeostasis being meticulously regulated, as this process is disrupted in cancer, with a notable alteration in protein composition and elevated secretion of proteases [17]. The tumor microenvironment (which consists of cells, ECM, soluble factors, signaling molecules, and mechanical cues) supports tumor growth and invasion, providing protection from host immunity and niches for dormant metastases to thrive as well as fostering therapeutic resistance [18]. The ECM, as one constituent of the multifaceted tumor microenvironment, undergoes significant alterations during the progression of breast tumorigenesis since cancer cells can dysregulate ECM remodeling, facilitating irreversible proteolysis and crosslinking and influencing cell signaling, angiogenesis, and tissue biomechanics and hence exerting profound effects on disease advancement [19,20].

The present review summarizes the current evidence regarding the intricate relationship between BC, ECM, and the microbiome. It discusses the emergence of this association and its future prospects.

## 2. Breast Cancer

BC arises from the uncontrolled proliferation of breast epithelial cells triggered by diverse carcinogenic factors. In advanced stages, cancer cells can metastasize to distant sites, forming multi-organ lesions that directly jeopardize affected individuals’ lives [21]. BC represents the most prevalent form of cancer, surpassing lung cancer [22], and is the leading cause of cancer-related mortality among females in most countries except Australia/New Zealand, Northern Europe, Northern America, and China, where lung cancer deaths prevail, posing a public health challenge [23,24]. BC is characterized by diverse clinical manifestations among women and has become as a prevalent malignancy [25], constituting 31% of all cases and 15% of projected fatalities attributed to women’s cancer in 2023 [26]. Annually, 2.3 million new BC cases are reported [27] across all genders and age cohorts [28]. Although BC is commonly associated with women, men can also be affected, albeit with a lower frequency. Male BC comprises approximately 1% of all BC cases [29]. Regarding BC survival, significantly discrepant rates are evident on a global scale. These rates range from approximately 80% for developed nations to below 40% for developing countries. These data highlight substantial disparities in healthcare access and resources [30]. Early diagnosis of BC can cure 70–80% of patients. Nevertheless, the prognosis for metastatic BC is dismal, constituting over 90% of BC-related mortality [31]. Despite advancements in treatment modalities for primary BCs (including surgery, chemotherapy, and radiotherapy), a significant portion of patients with distant metastases still experiences a lack of cure [32]. Despite the majority of patients being diagnosed with early-stage BC, up to 30% of these cases experience relapse within the initial 10 years of follow-up [33].

Most breast malignancies are adenocarcinomas. In general, breast carcinoma is divided into ductal carcinoma in situ (DCIS; a noninvasive potentially malignant intraductal proliferation of epithelial cells, confined to the ducts and lobules) and invasive ductal carcinoma (IDC; the most common form of invasive BC) [34]. Breast tumors are categorized by histology and immunochemistry [2]. The transition from phenotypic to intrinsic molecular BC subtypes has made a paradigm shift in BC treatment. This shift highlights the importance of individualizing therapy according to tumor biomarker status in the evolving regimens [35]. Based on American Society of Clinical Oncology guidelines [36], estrogen receptor (ER), progesterone receptor (PR), and human epidermal growth factor receptor 2 (HER2) are the recommended tumor marker tests for the prevention, screening, and treatment of BC. Gene expression profiling of breast carcinoma also allows to primarily classify in at least four distinct “intrinsic” subtypes, including two ER-positive (luminal A and luminal B) and two ER-negative (ERBB2 and basal subtypes) groups, with different expression patterns and characteristics among them [37]. Furthermore, the study of markers of elevated proliferation, which generally indicate a poor outcome, is of significant interest. In this line, Ki-67 antigen is used to evaluate BC proliferative activity and is also a prognostic biomarker of the risk of recurrence [38].

Tumorigenesis disrupts the mammary gland equilibrium and its environment, reprograming fibroblasts into a pro-invasive phenotype, which promotes ECM remodeling and the collective invasion of cancer cells [39]. In primary BC, tumor cells that resist T-cell attack are quiescent, forming clusters with reduced immune infiltration that display superior tumorigenic capacity. Thus, they constitute immunotherapy-resistant reservoirs by orchestrating a local hypoxic immune-suppressive milieu that blocks T-cell function [40]. Furthermore, during tumorigenesis, vascular networks are formed (a process known as angiogenesis) to nourish and supply oxygen to BC cells. Elevated levels of angiogenic factors are symptomatic of the aggressive nature of the respective tumor cells and correlate with a relatively poor prognosis [41]. During BC development, metastasis may occur, and the local disease can spread to lymph nodes or distant organs. These metastatic BC cells acquire aggressive characteristics from the tumor microenvironment through several mechanisms, including epithelial–mesenchymal transition (i.e., epithelial cells losing their cell polarity and adhesion properties and transforming into mesenchymal stem cells with enhanced migratory and invasive capabilities [42]) and epigenetic regulation [43].

## 3. Extracellular Matrix

### 3.1. Structure, Composition, and Molecular Aspects

The ECM is an intricate network composed of a matrix of multidomain macromolecules organized in a specific manner to form a structurally stable composite. This contributes to tissues’ mechanical properties [44,45]. The content and proportion of ECM components (i.e., fibrillar collagens, fibronectin, specific laminins, proteoglycans, and matricellular proteins) dictate its compliance, stiffness, porosity, viscoelasticity, and biochemical properties [45,46]. Among ECM proteins, there are large numbers of ECM-modifying enzymes, ECM-binding growth factors, and other ECM-associated proteins (collagen subunits, proteoglycans, and glycoproteins) [47]. The collagen superfamily of proteins plays a dominant role in maintaining the tissues’ integrity. Most collagens form polymeric assemblies such as fibrils, networks, and filaments. Some of them, such as endostatin, inhibit angiogenesis and tumor growth [48]. Moreover, basement membranes are layered cell-adherent ECM components that form part of the tissue architecture, serving as an extension of the plasma membrane, protecting tissues from disruptive physical stresses, and providing an interactive interface between cell and the surrounding environment [49].

### 3.2. ECM in the Cancer and Metastasis Context

The ECM is a highly dynamic entity that plays an active role in tumor progression due to its constituent components and mechanical characteristics [50]. Notably, the collagen-rich and laminin-rich basement membrane, along with the basement membrane stroma, serve as the boundary for tumor cells that metastasize only upon successful penetration through this basement membrane barrier [46]. Thus, the ECM serves as a regulator of the stem cell niche, with its stiffness playing a pivotal role in modulating cell growth, survival, tissue-specific differentiation, and tissue homeostasis, as tumors typically exhibit increased fibrotic and stiff ECM environments [51]. As a result of this shift in ECM homeostasis, interstitial fibrillar type I and III collagen as well as basement membrane protein type IV collagen are particularly relevant [17].

ECM remodeling, known as desmoplasia, characterized by the formation of excessive connective tissue around invasive carcinoma primarily but not exclusively of the breast [52], is orchestrated by myofibroblasts, also known as “activated fibroblasts”, which generate an ECM abundant in fibronectin and collagen, resulting in a fibrotic and stiff ECM milieu [53]. ECM degradation and epithelial–mesenchymal transition constitute the primary stages of local invasion, facilitated by the up-regulation of various proteinases, with a complex interplay of different signaling pathways [54,55]. Furthermore, endothelial cells are assembled into a laminin-rich basement membrane matrix, which is degraded during angiogenesis. As angiogenesis proceeds, the ECM serves essential functions in supporting key signaling events promoting endothelial cells’ migration, invasion, proliferation, and survival [56]. Hence, cancer cells can degrade the ECM and can proliferate and infiltrate surrounding tissues, a pivotal step in the dissemination of the primary tumor and the formation of metastases [57].

Metastatic disease results from the dissemination of metastatic tumor cells from the site of primary origin to distant organs. The tumor microenvironment is crucial to cancer progression, with various cell types participating in this process. These cell types include mesenchymal stem cells, lymphatic endothelial cells, cancer-associated fibroblasts, myeloid-derived suppressor cells, T cells, and tumor-associated macrophages [58]. The progression of metastatic carcinoma necessitates the initial penetration of the basement membrane, as neoplastic cells are compelled to enzymatically degrade the vascular sub-endothelial basement membrane [59]. During tissue remodeling, macrophages are recognized for their production of proteases (matrix metalloproteases and cathepsins) that facilitate the ECM breakdown, tumor cell migration, and invasion [60]. Additionally, cell adhesion to the ECM plays a pivotal role in regulating cellular processes (e.g., growth, differentiation, and apoptosis). However, cell–ECM adhesion mediated by integrins (specifically integrins α2 and α5) has been identified as a critical factor facilitating cancer cells’ adhesion, promoting metastasis and angiogenesis [61]. However, precise mechanisms related to cancer invasion and metastasis remain unknown due to their complexity [62]. The above-mentioned remodeling processes of ECM during cancer are summarized in Figure 1.

### 3.3. ECM-Targeting Therapies

ECM-targeting therapies have shown promise in various clinical applications, particularly in treating diseases where ECM dysregulation plays a significant role. For example, monoclonal antibodies [63] targeting ECM proteins (designed to interfere with the interactions between cancer cells and the ECM), small-molecule inhibitors of ECM-modifying enzymes [64] (target enzymes involved in the modification and turnover of ECM), matrix metalloproteinase (MMP) inhibitors [65] (which block their critical roles in the degradation and remodeling of the ECM), biological therapies targeting ECM-interacting proteins [66] (which involve using proteins, peptides, or antibodies to modulate interactions between cells and the ECM), and ECM-disrupting peptides [67] (short chains of amino acids designed to interfere with specific interactions between cells and the ECM).

### 3.4. Evidence Regarding ECM and BC

The ECM also plays a crucial role in normal breast function, such as mammary gland remodeling before and after lactation; its composition varies notably between metastatic and non-metastatic breast tumor stages [68]. Breast tumors are defined as epithelial neoplastic lesions distinguished by their dense structure, which predominantly comprises stromal cells and abundant ECM molecules. The reciprocal interactions between these components and epithelial cells are decisive in cancer progression [69]. In fact, BC progression frequently coincides with alterations in ECM stiffness and cellular adhesion capability, both intricately linked to cellular mechanotransduction [70]. Analyses of human BC samples have shown that tumor stiffness correlates with a more aggressive disease phenotype, as the progression is accompanied by substantial collagen deposition, linearization, and bundling [71]. Furthermore, the ECM plays a pivotal role in gene expression, cellular differentiation, and tissue organization during homeostasis. Hence, these factors have an essential role in the cellular invasion, metastasis, and drug responsiveness of BC cells [72].

During BC development, the ECM governs numerous pathways within cancer cells, including Wnt [73] (proteins that regulate cell-to-cell interactions as cell fate determination, cell proliferation, cell migration, and cell polarity), PI3K/AKT [74] (a critical intracellular signaling pathway that regulates various cellular processes, including metabolism, growth, proliferation, survival, and angiogenesis), ERK [75] (a series of phosphorylation events that culminate in the activation of ERK, whose dysregulation may lead to cancer, as it is a crucial intracellular signaling cascade), JNK [76] (involved in regulating inflammation, apoptosis, and stress responses that may be affected by stress stimuli, leading to the phosphorylation of target proteins and cancer), Src-FAK [77] (mediates cellular responses to extracellular signals, particularly those involving integrin-mediated adhesion and migration; Src kinase phosphorylates FAK at specific tyrosine residues, leading to the activation of downstream signaling pathways that regulate cytoskeletal dynamics, cell adhesion turnover, and cell motility), and Rho-GTPases [78] (essential for cellular functions such as the formation of stress fibers, regulation of cell polarity, and modulation of cell motility through their effects on actin cytoskeleton dynamics). Furthermore, the heightened deposition and crosslinking of collagens linked to tumorigenesis contributes to increased tissue stiffness [79]. Additionally, matrix metalloproteinases, which can enzymatically degrade diverse constituents of the ECM, and their functional genetic polymorphisms have been posited to potentially correlate with the BC risk [80]. Likewise, dysregulation in lysyl oxidase expression has been associated with the onset and progression of BC. This is due to the determinant role of this enzyme in the formation, maintenance, and functional characteristics of the ECM [81]. Another prominent ECM component within the stroma is hyaluronan (which belongs to the glycosaminoglycan family of polysaccharides), which play a significant role in the promotion of inflammation induced by BC [82].

Of significant importance, different inhibitors targeting various ECM components (e.g., integrins, proteoglycans, metalloproteinases, collagens, and the C1-peptidase protein families) have shown promising pre-clinical results in BC [68]. For instance, the inhibition of cathepsin B (namely the identified mutated CTSB gene in BC mice models) has been associated with reduced bone metastasis [83]. Likewise, the transforming growth factor β induced (TGFBI), which is released into the ECM (functioning mainly in cell adhesion, migration, proliferation, apoptosis, and angiogenesis), plays a suppressive role in the development of mesothelioma and BC cells, possibly through inhibitions of cell proliferation and inducing senescence [84]. The extracellular matrix protein 1 has also been proposed as a potential therapeutic target for overcoming tumor dissemination in BC metastasis. This is because of its apparent role in remodeling the actin cytoskeleton and in invadopodia (actin-based cortical protrusions of tumor cells required for stromal invasion and metastasis)[85].

## 4. Microbiome

### 4.1. The Human Microbiome

The human microbiome represents a complex and intricate ecological network, characterized by a diverse array of microorganisms (i.e., bacteria, viruses, fungi, and protozoa) that have established colonization at various anatomical sites throughout human bodies, such as the skin, oral cavity, vagina, or the gastrointestinal tract [86]. The microbiome inhabits and interacts with the human body. These interactions may be commensalistic, mutualistic, or pathogenic [87,88,89]. Furthermore, in recent years, the presence of microorganisms in organs or tissues has traditionally been considered as “sterile” [90]. These microbes have garnered increasing attention for their potential influence on diverse physiological processes (e.g., immune modulation, metabolism, and inflammation), and their role is emerging as a promising avenue for reshaping cancer development and therapy [91]. In this line, microbial dysbiosis has emerged as a crucial element in the development and progression of various cancers, including BC. A dysbiosis is defined as a derangement in bacterial abundance as a result of a decline in the richness and diversity of bacteria.

### 4.2. Microbiome and BC

Research evidence indicates a close relationship between microbial dysbiosis and BC [92,93]. BC manifests a notably diverse microbiome compared to normal breast samples and other forms of cancer. This diversity is marked by discernible variations across racial demographics, cancer stages, and BC subtypes [94,95]. These differences further exhibit variations based on clinicopathologic characteristics such as ER and PR status, levels of Ki-67, or HER2 status [96]. The characterization of the tumor microbiome remains challenging due to low biomass. Whether the presence of bacteria is advantageous to the tumors or to the bacteria themselves is unknown. For example, some BC subtypes characterized by increased oxidative stress are also enriched in bacteria that produce mycothiol, which detoxifies reactive oxygen species [97]. Thus, a thorough understanding of the breast microbiome and the impact of host genetics, lifestyle, and socioeconomic factors could be of significant interest in BC. The presence of some species has been found to be consistent across various studies, such as *Prevotella* and *Micrococcus* in normal breast tissue and *Lactobacillus* and *Fusobacterium* in malignant tumors of the breast [98,99,100,101]. In this line, it has been reported that bacterial composition differs from BC (where *Ralstonia* is abundant) to normal-breast individuals (highlighting the presence of *Acetobacter aceti*, *Lactobacillus vini*, *Lactobacillus paracasei*, and *Xanthomonas* spp.) [102]. Additionally, sequencing using breast tumor tissue and paired normal adjacent tissue from the same patient has shown that the bacterium *Methylobacterium radiotolerans* was relatively enriched in tumor tissue, while *Sphingomonas yanoikuyae* was enriched in the paired normal tissue, observing lower levels of antibacterial-response gene expression in tumor tissue [99]. Various processes, including the modulation of the immune response in BC, may be affected by the microbiome in breast tumors.

### 4.3. Oral Microbiome and BC

Recent research has underscored the significance of the oral microbiome, recognized as the most diverse microbiome within the human body, in the pathogenesis of various malignancies [103]. The current body of literature indicates a potential association between the oral microbiome and BC. Specifically, studies have demonstrated differences in oral microbial communities between BC patients and healthy women. BC risk is increased in women with periodontal disease. This is caused by bacteria such as *Porphyromonas gingivalis*, *Tannerella forsythia*, and *Treponema denticola* in the red complex. A number of other bacteria (*Prevotella intermedia*, *Fusobacterium nucleatum*, *Prevotella nigrescens*, *Streptococcus constellatus*, *Peptostreptococcus micros*, *Campylobacter showae*, *Eubacterium nodatum*, *Campylobacter gracilis*, and *Campylobacter rectus*) associated with periodontal disease have been identified in the orange complex [101,104,105]. BC cases showed lower relative abundances of *Porphyromonas* and *Fusobacterium* than controls. There is a strong correlation between alpha diversity and presence/relative abundance of specific genes from the oral and fecal microbiome among BC cases but not among controls. Among patients with BC, the relative abundance of oral *Porphyromonas* was inversely correlated with the relative abundance of fecal *Bacteroides* [106].

Moreover, one study identified a potential link between the oral microbiome, particularly that associated with menopausal and menstrual status, and BC risk [107]. This emerging association carries significant implications for cancer diagnostics, therapeutics, and prevention strategies. Consequently, comprehending the intricate interplay between the oral microbiome and carcinogenic pathways has emerged as a frontier in oncological research [108].

### 4.4. Gut Microbiome and BC

The importance of the gut microbiome to human health is increasingly recognized [109]. Gut dysbiosis has the potential to undermine gut barrier integrity, precipitating bacterial translocation and the subsequent establishment of a chronic inflammatory milieu within the gut. However, it remains unclear whether cancer pathology induces alterations in the gut microbiome or if dysbiosis itself serves as a carcinogenic factor [110]. Moreover, it is well established that chronic inflammation correlates with cancer growth and metastatic dissemination to distant organs [111]. In this regard, intracellular bacteria have been observed to accompany cancer cells within the circulatory system. This exerts a pivotal influence on tumor metastasis and colonization processes [112]. Obesity and a high-fat diet, particularly in cases of sporadic BC, may also be associated with the microbial communities residing within the host, potentially serving as a factor in breast carcinogenesis [113]. 

According to a cohort of 31 women with early BC, the abundance of *Blautia* species progressively increased with tumor grade, from 1.25 (1.01–1.43) in grade I to 2.95 (2.00–3.60) in grade III [114]. In patients with advanced clinical stages, *Bacteroidota*, *Clostridium coccoides*, and *Clostridium leptum* clusters were identified as well as *Faecalibacterium prausnitzii* and *Blautia* species [114].

Around 50% of breast carcinomas are associated with dietary factors. It is hypothesized that dietary fiber, by altering specific enzymatic activities such as β-glucuronidase, might influence the composition of the gut microbiome and estradiol metabolism, particularly among postmenopausal BC patients [115]. Furthermore, it has been reported that a reduction in gut bacterial diversity may contribute to estrogen release, ultimately resulting in an elevated BC risk [116]. Short-chain fatty acids such as butyrate can be produced in the gut microbiome through food digestion. These compounds possess enzyme-silencing properties that could potentially contribute to a decrease in BC development [117]. Additionally, low diversity in the gut microbiome has been correlated with diminished lymphocyte levels and elevated neutrophil counts, along with decreased survival rates among BC patients [118].

Given the gut microbiome’s capacity to influence estrogen metabolism and chronic inflammation linked to obesity, the composition of both the mammary and gut microbiome affects the risk of BC [119]. It has also been observed that restoring gut microbial homeostasis represents an emerging therapeutic strategy in BC treatment. Certain probiotics have demonstrated potential for preventing or treating BC by modulating gastrointestinal bacteria and the systemic immune system [120]. Another crucial element to consider is the role of phytoestrogens, which are metabolized by the gut microbiome and exert their effects through interaction with ERs, particularly ERβ. This interaction modulates target tissue responses. Although phytoestrogens typically exhibit lower receptor binding affinity than endogenous estrogen, their biological effects are contingent upon circulating estrogen levels, leading to context-dependent estrogenic or antiestrogenic actions [121]. Moreover, the gut microbiome influences chemotherapy efficacy and toxicity, thereby affecting treatment outcomes [122]. 

It must be noted that the gut microbiome is not exclusively shaped by dietary intake but also influenced by medications; certain studies indicate correlations between the microbiome composition and the response to chemotherapy, radiotherapy, and immunotherapy. Hence, interventions targeting microbiome manipulation could potentially be developed to prevent or treat BCs, optimizing therapeutic outcomes and mitigating adverse effects [123].

## 5. Relationship between BC, ECM, and the Microbiome

As part of its role in health maintenance, the ECM controls the diffusion of infections and inflammations, detects and adapts to external stimuli from the environment, and, in the case of cancer, interacts with cancer cells and controls tumor invasion and drug resistance [124]. Around BC masses, an accumulation of type I collagen fibrils, i.e., desmoplasia, occurs in order to resist cancer invasion mechanically [124]. Later, these same fibrils bundle to form fibers that run parallel to the surface of the tumor and prevent cancer cells from invading and spreading [124]. Cancer niche formation, tumor progression, and drug resistance are associated with positive interactions between tumor cells and the microbiome. In addition, several enzymes have been identified that are capable of degrading the host ECM (e.g., collagenase, hyaluronidase (HAase) and elastase) [125,126].

Hyaluronic acid is a major matrix molecule expressed in human malignancies [127]. There is a significant overexpression of it in BC (approximately 56%) [128]. This results in an increased biophysical barrier that significantly compresses blood vessels, increases interstitial fluid pressure, and hinders drug delivery [129,130]. Inflammation and cancer progression are facilitated by the binding of hyaluronic acid to the transmembrane receptor CD44, which is necessary for cell migration, proliferation, and invasion [131,132]. In clinical applications, especially oncology, hyaluronidase facilitates the dispersion and absorption of interstitial fluids. HAase protein derived from *Staphylococcus aureus* ATCC 29213 exhibits higher enzyme activity than that derived from other strains [133]. By degrading HA in the ECM, HAase reduces the interstitial fluid pressure in the tumor ECM. This enhances chemotherapy by activating interstitial space dispersion [134].

As previously mentioned, activated fibroblasts or myofibroblasts are responsible for the synthesis and remodeling of the ECM, which provides a path for tumor cells to exit the primary tissue and disseminate throughout the body [135]. Commensal dysbiosis increases the frequency of mast cells in the mammary tissue as well as their profibrogenicity, a phenotypic change that persists after tumor implantation [136]. The activation of fibroblasts and remodeling of tissue are associated with enhanced breast tumor metastasis. Through pharmacological and adoptive transfer approaches, Feng et al. were able to demonstrate that mast cells from dysbiotic animals increase the dissemination of hormone receptor-positive tumor cells in mammary tissue. Mammary collagen levels were correlated with mast cell abundance in hormone receptor-positive BC patients, suggesting that mast cell-mediated fibroblast activation may have clinical relevance. In conjunction, these data indicate that fibroblast activation and early tumor dissemination are mediated by a gut–mast cell axis [136].

Invasion of BC requires the stiffening and self-assembly of the matrix made up of collagen I and ECM modifiers [137]. A pathologic ECM promotes BC cell proliferation by activating the FAK and ERK pathways [138]. In BC, the balance between intracellular and cell–matrix adhesions determines the ECM dynamics and the invasion of cancerous cells [139]. Inflammation and remodeling of ECM have been reported in association with *Anaerococcus* [140]. A key role may be played by the interaction between the ECM, the microbiome, and inflammation in the onset, progression, and relapse of BC [141]. In general, BC patients showed an increase in bacterial richness [142]. According to Zeng et al. [143], patients with low alpha diversity (variation of microbes in a single sample) have a significantly longer recurrence-free survival than those with high alpha diversity [143]. 

## 6. Further Perspectives

BC molecular subtyping is crucial for predictive and prognostic purposes due to diverse clinical behaviors observed across types. Recently, different alternatives and novel biomarkers of BC identification have emerged that may improve clinical outcomes and provide tailored treatments.

In this context, radiogenomics has the potential to significantly improve early detection, prognosis, and diagnosis of BC since it allows for visual features and genetic marker linkage that promise to eliminate the need for biopsy and sequencing [144]. Similarly, liquid biopsy using ctDNA, which has been reported to be less invasive and effective for comprehensive genetic analysis of heterogeneous solid tumors, could be useful across a wide range of cancer types, including BC. Its clinical applications are expected to expand further through ongoing research [145]. Likewise, the application of circRNAs in clinical practice for diagnosing, treating, and monitoring BC could be another promising alternative. Nevertheless, the interaction and influence of multiple pathways must be investigated, requiring more in-depth studies to improve the understanding of the molecular mechanism of the circRNA network [146]. Additionally, it has been highlighted that the potential diagnostic function of lncRNA H-19 and miR-200a in BC as well as the association of IL-6/SIRT-1 with lncRNA H-19/miR-200a expression could be a promising opportunity for clinical outcomes and tailored treatments [147]. Lastly, the levels of advanced oxidation protein products appear to have a significant role in predicting cancer-related events and may potentially serve as a simple prognostic marker in clinical practice [148].

Several treatment options for BC are available today, including surgery, chemotherapy, radiotherapy and immunotherapy, the last being a highly specific therapy involving adaptive immune responses and immunological memory. In this field, research has proposed the creation of cancer vaccines and immunotherapies for BC malignancies, targeting proteins such as the GRP78 protein [149]. Immunotherapy appears to have widely different results, with some patients showing a pronounced clinical response, while others experience minimal or no clinical benefit from the same treatment. Hence, understanding the complexity and diversity of the immune context of the tumor microenvironment could predict and guide the immunotherapeutic response [150]. On the other hand, bacterial therapeutics for tumor treatment and immune modulation are another promising alternative. This is due to the significance of the microbiome in mitigating BC via its anti-tumor activities [151]. Microbiome composition categorizes BC patients into subgroups, thereby facilitating tailored treatment approaches with enhanced efficacy. The integration of cell lines and circulating tumor cells with advanced analytical techniques unveils intricate mechanisms [152]. However, bacteriotherapy, aimed at reversing gut dysbiosis and enhancing diagnostic outcomes, is not without challenges [151]. 

The results of studies in tumor mouse models of BC (4T1) showed that HAase-mediated ECM degradation reduces interstitial fluid pressure in the tumor microenvironment, suggesting that this strategy may overcome the limitations of chemotherapy. By using oncolytic bacteria to target and degrade ECM, drug penetration management will be facilitated in the future, thereby improving the outcomes of various oncology therapies, particularly in BC [134].

In summary, although the significance of the microbiome and other therapeutic approaches in BC pathogenesis is garnering escalating attention, it is imperative to acknowledge the inherent challenges, which encompass, among others, disparities in sample collection techniques, variations in DNA extraction methodologies, potential issues of contamination, and the necessity for rigorous bioinformatics and statistical analyses to interpret intricate datasets [153]. A summary of the main information presented in this review is graphically represented in Figure 2.

## Figures and Tables

**Figure 1 ijms-25-07226-f001:**
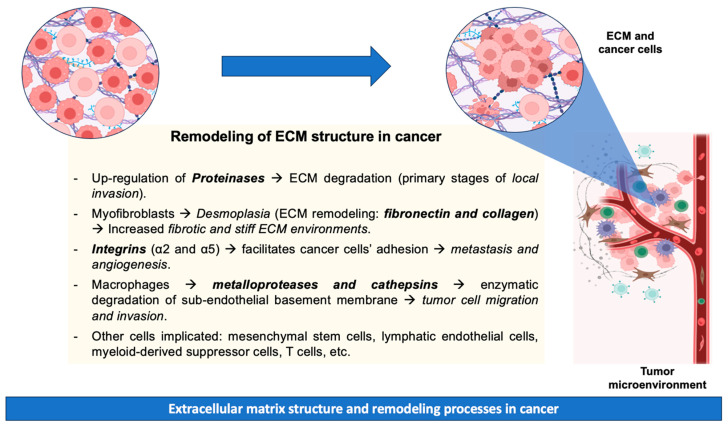
Remodeling processes of ECM during cancer.

**Figure 2 ijms-25-07226-f002:**
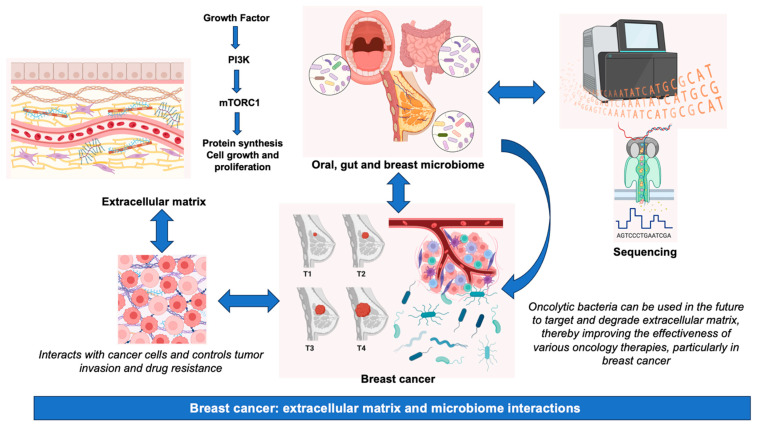
Breast cancer: extracellular matrix and microbiome interactions. Abbreviations, mTORC1, mammalian target of rapamycin complex 1; PI3K, phosphatidylinositol-3 kinase.

## Data Availability

Not applicable.

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
