# Peer review of "Breast Cancer: Extracellular Matrix and Microbiome Interactions"

_ijms, 2024, doi:10.3390/ijms25137226_

Round 1

Reviewer 1 Report

Comments and Suggestions for Authors

This review article aims to summarize the findings arrived from studies regarding the intricate relationship between breast cancer with the extracellular matrix and with microbiome. This is a rather new field of research, however there is a lot of information that it needs to be collected and reviewed for educational and research purposes.

The authors have well-organized the article, with a general introductory chapter on cancer and microbiome, followed by a detailed chapter on breast cancer, a more detailed chapter on extracellular matrix and an even more detailed chapter on microbiome. A chapter summarizing the relationship between breast cancer- extracellular matrix-microbiome follows, and the review article ends with the future perspectives.

It is suggested to the authors to revise the sentence “BC represents the most prevalent form of cancer, surpassing lung cancer [16] and being the leading cause of cancer-related mortality among females worldwide, posing a public health challenge [17]” to “BC represents the most prevalent form of cancer, surpassing lung cancer [16] and being the leading cause of cancer-related mortality among females in most of the countries, except Australia/New Zealand, Northern Europe, Northern America, and China, where lung cancer deaths prevail, posing a public health challenge”, as it has been presented in https://doi.org/10.3322/caac.21660.

Author Response

June 25th, 2024

Albert Li (Section Managing Editor)

Dear Mr. Albert Li,

Thank you for providing us the opportunity to submit a revised version of our manuscript entitled “Breast cancer: extracellular matrix and microbiome interactions” to International Journal of Molecular Sciences. The authors thank the reviewers for their thoughtful comments and suggestions on our manuscript. We have considered all of the comments and incorporated them into the revised manuscript. Changes to the original document are highlighted as track changes, and an itemized point-by-point response to the reviewers’ comments is presented below.

COMMENTS FROM REVIEWER #1

This review article aims to summarize the findings arrived from studies regarding the intricate relationship between breast cancer with the extracellular matrix and with microbiome. This is a rather new field of research, however there is a lot of information that it needs to be collected and reviewed for educational and research purposes. The authors have well-organized the article, with a general introductory chapter on cancer and microbiome, followed by a detailed chapter on breast cancer, a more detailed chapter on extracellular matrix and an even more detailed chapter on microbiome. A chapter summarizing the relationship between breast cancer- extracellular matrix-microbiome follows, and the review article ends with the future perspectives.

Response: We are grateful for the reviewer's kind and helpful comments regarding our manuscript.

It is suggested to the authors to revise the sentence “BC represents the most prevalent form of cancer, surpassing lung cancer [16] and being the leading cause of cancer-related mortality among females worldwide, posing a public health challenge [17]” to “BC represents the most prevalent form of cancer, surpassing lung cancer [16] and being the leading cause of cancer-related mortality among females in most of the countries, except Australia/New Zealand, Northern Europe, Northern America, and China, where lung cancer deaths prevail, posing a public health challenge”, as it has been presented in https://doi.org/10.3322/caac.21660.

Response: Dear reviewer, as suggested, we have revised and reformulated the sentence and now state (page 2, lines 79-82), “BC represents the most prevalent form of cancer, surpassing lung cancer [22] and being the leading cause of cancer-related mortality among females in most of the countries, except Australia/New Zealand, Northern Europe, Northern America, and China, where lung cancer deaths prevail, posing a public health challenge [23,24].

Reviewer 2 Report

Comments and Suggestions for Authors

The manuscript is about the correlation between the microbiome and breast cancer risk. The review is organized into several subsections, which include a description of breast cancer, extracellular matrix, and microbiome. The separate section is about the relationship between breast cancer, ECM, and the microbiome.  There is increasing data on the role of the microbiome in health and disease. Therefore, the topic of review is very actual. I have a few comments. In lane 36, I propose to use reproductive factors instead of age-related factors. In lane 49, I think that it should be used consist of, but not „consist in). In lane 203, I think that „transforming” is missing in the name of growth factor β in lane 203.

Author Response

June 25th, 2024

Albert Li (Section Managing Editor)

Dear Mr. Albert Li,

Thank you for providing us the opportunity to submit a revised version of our manuscript entitled “Breast cancer: extracellular matrix and microbiome interactions” to International Journal of Molecular Sciences. The authors thank the reviewers for their thoughtful comments and suggestions on our manuscript. We have considered all of the comments and incorporated them into the revised manuscript. Changes to the original document are highlighted as track changes, and an itemized point-by-point response to the reviewers’ comments is presented below.

COMMENTS FROM REVIEWER #2

The manuscript is about the correlation between the microbiome and breast cancer risk. The review is organized into several subsections, which include a description of breast cancer, extracellular matrix, and microbiome. The separate section is about the relationship between breast cancer, ECM, and the microbiome.  There is increasing data on the role of the microbiome in health and disease. Therefore, the topic of review is very actual.

Response: We appreciate the kind and helpful comments that the reviewer provided regarding our manuscript.

I have a few comments. In line 36, I propose to use reproductive factors instead of age-related factors.

Response: The sentence has been rewritten as suggested

In line 49, I think that it should be used consist of, but not „consist in).

Response: We thank the reviewer suggestion. The term “of” was more suitable instead of “in”.

In line 203, I think that „transforming” is missing in the name of growth factor β in lane 203.

Response: We did not include transforming as proposed. We thank the reviewer suggestion which has been included.

Reviewer 3 Report

Comments and Suggestions for Authors

Dear authors,

This review offers valuable insights into the intricate relationships between the ECM, microbiome, and BC, paving the way for future research and therapeutic innovations. All sections, the abstract, introduction, and subsections are clear and well-structured, providing a logical flow from the significance of breast cancer to the specific focus on ECM and microbiome. The content is sufficiently detailed to inform readers about the current state of breast cancer research, risk factors, treatment challenges, and the emerging roles of ECM and microbiome. The emphasis on ECM and microbiome is relevant given the growing interest in these areas for cancer research and treatment development. Minor improvements in the clarity of the introduction, as well as a more detailed discussion of therapeutic targets, could enhance the manuscript further.

Recommendations for improvement of introduction:

Comment 1: While the introduction mentions various risk factors and treatment limitations, it could benefit from more specific examples or recent data to illustrate these points.

Comment 2: The relationship between ECM remodeling and the microbiome in the context of breast cancer could be introduced earlier in the introduction to create a more cohesive narrative.

Recommendations for improvement in subsections:

Comment 3: The authors should expand the explanation of pathways like Wnt, PI3K/AKT, and others to clarify their roles in ECM-related BC progression.

Comment 4: The authors could incorporate diagrams or charts illustrating ECM structure and remodeling processes in cancer to enhance understanding.

Comment 5: The authors should define technical terms (e.g., "desmoplasia," "epithelial-mesenchymal transition") to ensure clarity for readers less familiar with the subject.

Comment 6: The authors should provide more detailed examples of ECM-targeting therapies and their mechanisms to illustrate potential clinical applications better.

Author Response

June 25th, 2024

Albert Li (Section Managing Editor)

Dear Mr. Albert Li,

Thank you for providing us the opportunity to submit a revised version of our manuscript entitled “Breast cancer: extracellular matrix and microbiome interactions” to International Journal of Molecular Sciences. The authors thank the reviewers for their thoughtful comments and suggestions on our manuscript. We have considered all of the comments and incorporated them into the revised manuscript. Changes to the original document are highlighted as track changes, and an itemized point-by-point response to the reviewers’ comments is presented below.

COMMENTS FROM REVIEWER #3

Dear authors,

This review offers valuable insights into the intricate relationships between the ECM, microbiome, and BC, paving the way for future research and therapeutic innovations. All sections, the abstract, introduction, and subsections are clear and well-structured, providing a logical flow from the significance of breast cancer to the specific focus on ECM and microbiome. The content is sufficiently detailed to inform readers about the current state of breast cancer research, risk factors, treatment challenges, and the emerging roles of ECM and microbiome. The emphasis on ECM and microbiome is relevant given the growing interest in these areas for cancer research and treatment development. Minor improvements in the clarity of the introduction, as well as a more detailed discussion of therapeutic targets, could enhance the manuscript further.

Response: It was a pleasure to receive the helpful and kind comments from the reviewer regarding our manuscript.

Recommendations for improvement of introduction:

Comment 1: While the introduction mentions various risk factors and treatment limitations, it could benefit from more specific examples or recent data to illustrate these points.

Response: We have detailed the novel and most recent approaches for breast cancer in the introduction section as suggested, and now state (pages 1-2, lines 43-51), “Although conventional treatments such as chemotherapy, surgery, hormone therapy or radiotherapy are commonly used, they often entail significant side effects. Therefore, there is a critical need to investigate more cost-effective and efficient treatment modalities for BC [7] like personalized medicine [8] based on genetic, phenotypic, and environ-mental factors, minimally invasive procedures [9] as laparoscopic and robotic surgery, improved screening and early detection [10], cost-effective [11] drug development, combination therapies [12] using combinations of drugs that work synergistically, and telemedicine and remote monitoring [13].”

Comment 2: The relationship between ECM remodeling and the microbiome in the context of breast cancer could be introduced earlier in the introduction to create a more cohesive narrative,

Response: As suggested, the relationship between ECM remodeling and microbiome has been introduced earlier for clarifying the introduction section and now state (page 2, lines 52-60), “A novel factor which has emerged in recent years which could be linked with BC is the microbiome, which has been identified as a fundamental contributor to human health and overall wellbeing [14]. It can exert influence on a multitude of human biological, hormonal, and metabolic pathways. This influence can potentially stimulate the initiation, proliferation, and genetic instability of cancer within host cells, or alternatively, induce apoptosis [15]. The interplay between tumor cells and their microenvironment has been recognized as one of the key determinants of cancer development – data suggest that both the ECM and the microbiome represent microenvironments that contribute to the onset and progression of tumors [16].”

Recommendations for improvement in subsections:

Comment 3: The authors should expand the explanation of pathways like Wnt, PI3K/AKT, and others to clarify their roles in ECM-related BC progression.

Response: We have expanded all the complex pathways for a better understanding (i.e., Wnt, PI3K/AKT, ERK, JNK, Src-FAK, Rho-GTPases). We thank the reviewer for his suggestion. The manuscript now state (page 5, lines 239-253), “During BC development, the ECM governs numerous pathways within cancer cells, including Wnt [73](proteins that regulate cell-to-cell interactions as cell fate determination, cell proliferation, cell migration, and cell polarity), PI3K/AKT [74](a critical intracellular signaling pathway that regulates various cellular processes, including metabolism, growth, proliferation, survival, and angiogenesis), ERK [75](a series of phosphorylation events that culminate in the activation of ERK, whose dysregulation may lead to cancer as it is a crucial intracellular signaling cascade), JNK [76](involved in regulating inflammation, apoptosis, and stress responses may be affected by stress stimuli, leading to the phosphorylation of target proteins and cancer), Src-FAK [77](mediates cellular responses to extracellular signals, particularly those involving integrin-mediated adhesion and migration. Src kinase phosphorylates FAK at specific tyrosine residues, leading to the activation of downstream signaling pathways that regulate cytoskeletal dynamics, cell adhesion turnover, and cell motility), and Rho-GTPases [78] (essential for cellular functions such as the formation of stress fibers, regulation of cell polarity, and modulation of cell motility through their effects on actin cytoskeleton dynamics).”

Comment 4: The authors could incorporate diagrams or charts illustrating ECM structure and remodeling processes in cancer to enhance understanding.

Response: Dear reviewer, we have created a new figure (figure 1 in the manuscript) summarizing the main processes implicated in remodeling ECM structure during cancer.

Comment 5: The authors should define technical terms (e.g., "desmoplasia," "epithelial-mesenchymal transition") to ensure clarity for readers less familiar with the subject.

Response: The above-referred terms have been explained for a better understanding. Comment appreciated.

Comment 6: The authors should provide more detailed examples of ECM-targeting therapies and their mechanisms to illustrate potential clinical applications better.

Response: We have created a subsection named 3.3. ECM-targeting therapies in which multiple targeting therapies have been mentioned and developed. We thank the reviewer suggestion. The manuscript now state (page 4, lines 209-220), “3.3. ECM-targeting therapies. ECM-targeting therapies have shown promise in various clinical applications, particularly in treating diseases where ECM dysregulation plays a significant role. For example, monoclonal antibodies [63] targeting ECM proteins (designed to interfere with the interactions between cancer cells and the ECM), small molecule inhibitors of ECM-modifying enzymes [64](target enzymes involved in the modification and turnover of ECM), matrix metalloproteinase (MMP) inhibitors [65](block their critical roles in the degradation and remodeling of the ECM), biological therapies targeting ECM-interacting proteins [66] (involve using proteins, peptides, or antibodies to modulate interactions between cells and the ECM), and ECM-disrupting peptides [67] (short chains of amino acids designed to interfere with specific interactions between cells and the ECM).”.
